# Hyperparameter Learning via Distributional Transfer

**Ho Chung Leon Law**[*]
University of Oxford
ho.law@stats.ox.ac.uk

**Peilin Zhao**[*]
Tencent AI Lab
masonzhao@tencent.com

**Lucian Chan**
University of Oxford
leung.chan@stats.ox.ac.uk

**Junzhou Huang**
Tencent AI Lab
joehhuang@tencent.com

**Dino Sejdinovic**[*]
University of Oxford
dino.sejdinovic@stats.ox.ac.uk

## Abstract

Bayesian optimisation is a popular technique for hyperparameter learning but typically requires initial exploration even in cases where similar prior tasks have been solved. We propose to transfer information across tasks using learnt representations of training datasets used in those tasks. This results in a joint Gaussian process model on hyperparameters and data representations. Representations make use of the framework of distribution embeddings into reproducing kernel Hilbert spaces. The developed method has a faster convergence compared to existing baselines, in some cases requiring only a few evaluations of the target objective.

## 1 Introduction

Hyperparameter selection is an essential part of training a machine learning model and a judicious choice of values of hyperparameters such as learning rate, regularisation, or kernel parameters is what often makes the difference between an effective and a useless model. To tackle the challenge in a more principled way, the machine learning community has been increasingly focusing on Bayesian optimisation (BO) [34], a sequential strategy to select hyperparameters $\theta$ based on past evaluations of model performance. In particular, a Gaussian process (GP) [31] prior is used to represent the underlying accuracy $f$ as a function of the hyperparameters $\theta$, whilst different acquisition functions $\alpha(\theta; f)$ are proposed to balance between exploration and exploitation. This has been shown to give superior performance compared to traditional methods [34] such as grid search or random search. However, BO suffers from the so called 'cold start' problem [28, 38], namely, initial observations of $f$ at different hyperparameters are required to fit a GP model. Various methods [38, 6, 36, 28] were proposed to address this issue by transferring knowledge from previously solved tasks, however, initial random evaluations of the models are still needed to consider the similarity across tasks. This might be prohibitive: evaluations of $f$ can be computationally costly and our goal may be to select hyperparameters and deploy our model as soon as possible. We note that treating $f$ as a black-box function, as is often the case in BO, is ignoring the highly structured nature of hyperparameter learning – it corresponds to training specific models on specific datasets. We make steps towards utilizing such structure in order to borrow strength across different tasks and datasets.

**Contribution.** We consider a scenario where a number of tasks have been previously solved and we propose a new BO algorithm, making use of the embeddings of the distribution of the training data

---

[*]Corresponding authors

[4, 23]. In particular, we propose a model that can jointly model all tasks at once, by considering an extended domain of inputs to model accuracy $f$, namely the distribution of the training data $\mathcal{P}_{XY}$, sample size of the training data $s$ and hyperparameters $\theta$. Through utilising *all* seen evaluations from all tasks and meta-information, our methodology is able to learn a useful representation of the task that enables appropriate transfer of information to new tasks. As part of our contribution, we adapt our modelling approach to recent advances in scalable hyperparameter transfer learning [26] and demonstrate that our proposed methodology can scale linearly in the number of function evaluations. Empirically, across a range of regression and classification tasks, our methodology performs favourably at initialisation and has a faster convergence compared to existing baselines – in some cases, the optimal accuracy is achieved in just a few evaluations.

## 2   Related Work

The idea of transferring information from different tasks in the context of hyperparameter learning has been studied in various settings [38, 6, 36, 28, 43, 26]. Amongst this literature, one common feature is that the similarity across tasks is captured only through the evaluations of $f$. This implies that sufficient evaluations from the task of interest is *necessary*, before we can transfer information. This is problematic, if model training is computationally expensive and our goal is to employ our model as quickly as possible. Further, the hyperparameter search for a machine learning model in general is not a black-box function, as we have additional information available: the dataset used in training. In our work, we aim to learn feature representation of training datasets in-order to yield good initial hyperparameter candidates without having seen any evaluations from our target task.

While such use of such dataset features, called *meta-features*, has been previously explored, current literature focuses on handcrafted meta-features[2]. These strategies are not optimal, as these meta-features can be be very similar, while having very different $f$s, and vice versa. In fact a study on OpenML [40] meta-features have shown that the optimal set depends on the algorithm and data [39]. This suggests that the reliance on these features can have an adverse effect on exploration, and we give an example of this in section 5. To avoid such shortcomings, given the same input space, our algorithm is able to *learn* meta-features directly from the data, avoiding such potential issues.

Although [15] previously have also proposed to learn the meta-feature representations (for image data specifically), their proposed methodology requires the same set of hyperparameters to be evaluated for all previous tasks. This is clearly a limitation considering that different hyperparameter regions will be of interest for different tasks, and we would thus require excessive exploration of all those different regions under each task. To utilise meta-features, [15] propose to warm-start Bayesian optimisation [10, 32, 8] by initialising with the best hyperparameters from previous tasks. This also might be sub-optimal as we neglect non-optimal hyperparameters that can still provide valuable information for our new task, as we demonstrate in section 5. Our work can be thought of to be similar in spirit to [17], which considers an additional input to be the sample size $s$, but do not consider different tasks corresponding to different training data distributions.

## 3   Background

Our goal is to find:
$$\theta^*_{\text{target}} = \text{argmax}_{\theta \in \Theta} f^{\text{target}}(\theta)$$
where $f^{\text{target}}$ is the target task objective we would like to optimise with respect to hyperparameters $\theta$. In our setting, we assume that there are $n$ (potentially) related source tasks $f^i, i = 1, \ldots n$, and for each $f^i$, we assume that we have $\{\theta^i_k, z^i_k\}_{k=1}^{N_i}$ from past runs, where $z^i_k$ denotes a noisy evaluation of $f^i(\theta^i_k)$ and $N_i$ denotes the number of evaluations of $f^i$ from task $i$. Here, we focus on the case that $f^i(\theta)$ is some standardised accuracy (e.g. test set AUC) of a trained machine learning model with hyperparameters $\theta$ and training data $D_i = \{\mathbf{x}^i_\ell, y^i_\ell\}_{\ell=1}^{s_i}$, where $\mathbf{x}^i_\ell \in \mathbb{R}^p$ are the covariates, $y^i_\ell$ are the labels and $s_i$ is the sample size of the training data. For a general framework, $D_i$ is any input to $f^i$ apart from $\theta$ (can be unsupervised) – but following a typical supervised learning treatment, we assume it to be an i.i.d. sample from the joint distribution $\mathcal{P}_{XY}$. For each task we now have:
$$(f^i, D_i = \{\mathbf{x}^i_\ell, y^i_\ell\}_{\ell=1}^{s_i}, \{\theta^i_k, z^i_k\}_{k=1}^{N_i}), \quad i = 1, \ldots n$$

Our strategy now is to measure the similarity between datasets (as a representation of the task itself), in order to transfer information from previous tasks to help us quickly locate $\theta_{\text{target}}^*$. In order to construct meaningful representations and measure between different tasks, we will make the assumption that $\mathbf{x}_\ell^i \in \mathcal{X}$ and $y_\ell^i \in \mathcal{Y}$ for all $i$, and that throughout the supervised learning model class is the same. While this setting might seem limiting, there are many examples of practical applications, including ride-sharing, customer analytics model and online inventory system [6, 28]. In all these cases, as new data becomes available, we might want to either re-train our model or re-fit our parameters of the system to adapt to a specific distributional data input. In section 5.3, we further demonstrate that our methodology is applicable to a real life protein-ligand binding problem in the area of drug design, which typically require significant efforts to tune hyperparameters of the models for different targets [33].

Intuitively, this assumption implies that the source of differences of $f^i(\theta)$ across $i$ and $f^{\text{target}}(\theta)$ is in the data $D_i$ and $D_{\text{target}}$. To model this, we will decompose the data $D_i$ into the joint distribution $\mathcal{P}_{XY}^i$ of the training data ($D_i = \{\mathbf{x}_\ell^i, y_\ell^i\}_{\ell=1}^{s_i} \overset{i.i.d.}{\sim} \mathcal{P}_{XY}^i$) and the sample size $s_i$ for task $i$. Sample size[3] is important here as it is closely related to model complexity choice which is in turn closely related to hyperparameter choice [17]. While we have chosen to model $D_i$ as $P_{XY}^i$ and $s_i$, in practice through simple modifications of the methodology we propose, it is possible to model $D_i$ as a set [44]. Under this setting, we will consider $f(\theta, \mathcal{P}_{XY}, s)$, where $f$ is a function on hyperparameters $\theta$, joint distribution $\mathcal{P}_{XY}$ and sample size $s$. For example, $f$ could be the negative empirical risk, i.e.

$$f(\theta, \mathcal{P}_{XY}, s) = -\frac{1}{s}\sum_{\ell=1}^{s} L(h_\theta(\mathbf{x}_\ell), y_\ell),$$

where $L$ is the loss function and $h_\theta$ is the model's predictor. To recover $f^i$ and $f^{\text{target}}$, we can evaluate at the corresponding $\mathcal{P}_{XY}$ and $s$, i.e. $f^i(\theta) = f(\theta, \mathcal{P}_{XY}^i, s_i)$, $f^{\text{target}}(\theta) = f(\theta, \mathcal{P}_{XY}^{\text{target}}, s_{\text{target}})$. In this form, we can see that similarly to assuming that $f$ varies smoothly as a function of $\theta$ in standard BO, this model also assumes smoothness of $f$ across $\mathcal{P}_{XY}$ as well as across $s$ following [17]. Here we can see that if two distributions and sample sizes are similar (with respect to a distance of their representations that we will learn), their corresponding values of $f$ will also be similar. In this source and target task setup, this would suggest we can selectively utilise information from previous source datasets evaluations $\{\theta_k^i, z_k^i\}_{k=1}^{N_i}$ to help us model $f^{\text{target}}$.

# 4 Methodology

## 4.1 Embedding of data distributions

To model $\mathcal{P}_{XY}$, we will construct $\psi(D)$, a feature map on joint distributions for each task, estimated through its task's training data $D$. Here, we will follow [4] which considers transfer learning, and make use of kernel mean embedding to compute feature maps of distributions (cf. [23] for an overview). We begin by considering various feature maps of covariates and labels, denoting them by $\phi_x(\mathbf{x}) \in \mathbb{R}^a$, $\phi_y(y) \in \mathbb{R}^b$ and $\phi_{xy}([\mathbf{x}, y]) \in \mathbb{R}^c$, where $[\mathbf{x}, y]$ denotes the concatenation of covariates $\mathbf{x}$ and label $y$. Depending on the different scenarios, different quantities will be of interest.

**Marginal Distribution** $P_X$. Modelling of the marginal distribution $P_X$ is useful, as we might expect various tasks to differ in the distribution of $\mathbf{x}$ and hence in the hyperparameters $\theta$, which, for example, may be related to the scales of covariates. We also might find that $\mathbf{x}$ is observed with different levels of noise across tasks. In this situation, it is natural to expect that those tasks with more noise would perform better under a simpler, more robust model (e.g. by increasing $\ell_2$ regularisation in the objective function). To embed $P_X$, we can estimate the kernel mean embedding $\mu_{P_X}$ [23] with $D$ by:

$$\psi(D) = \hat{\mu}_{P_X} = \frac{1}{s}\sum_{\ell=1}^{s} \phi_x(\mathbf{x}_\ell)$$

where $\psi(D) \in \mathbb{R}^a$ is an estimator of a representation of the marginal distribution $P_X$.

**Conditional Distribution** $P_{Y|X}$. Similar to $P_X$, we can also embed the conditional distribution $P_{Y|X}$. This is an important quantity, as across tasks, the form of the signal can shift. For example, we

might have a latent variable $W$ that controls the smoothness of a function, i.e. $P^i_{Y|X} = P_{Y|X,W=w_i}$. In a ridge regression setting, we will observe that those tasks (functions) that are less smooth would require a smaller bandwidth $\sigma$ in order to perform better. For regression, to model the conditional distribution, we will use the kernel conditional mean operator $C_{Y|X}$ [35] estimated with $D$ by:

$$\hat{\mathcal{C}}_{Y|X} = \Phi_y^\top(\Phi_x\Phi_x^\top + \lambda I)^{-1}\Phi_x = \lambda^{-1}\Phi_y^\top(I - \Phi_x(\lambda I + \Phi_x^\top\Phi_x)^{-1}\Phi_x^\top)\Phi_x$$

where $\Phi_x = [\phi_x(\mathbf{x}_1), \ldots, \phi_x(\mathbf{x}_s)]^T \in \mathbb{R}^{s\times a}$, $\Phi_y = [\phi_y(y_1), \ldots, \phi_y(y_s)]^T \in \mathbb{R}^{s\times b}$ and $\lambda$ is a regularisation parameter that we learn. It should be noted the second equality [31] here allows us to avoid the $O(s^3)$ arising from the inverse. This is important, as the number of samples $s$ per task can be large. As $\hat{\mathcal{C}}_{Y|X} \in \mathbb{R}^{b\times a}$, we will flatten it to obtain $\psi(D) \in \mathbb{R}^{ab}$ to obtain a representation of $P_{Y|X}$. In practice, as we rarely have prior insights into which quantity is useful for transferring hyperparameter information, we will model both the marginal and conditional distributions together by concatenating the two feature maps above. The advantage of such an approach is that the learning algorithm does not have to itself decouple the overall representation of training dataset into the information about marginal and conditional distributions which is likely to be informative.

**Joint Distribution** $P_{XY}$. Taking an alternative and a more simplistic approach, it is also possible to model the joint distribution $P_{XY}$ directly. One approach is to compute the kernel mean embedding, based on concatenated samples $[\mathbf{x}, y]$, considering the feature map $\phi_{xy}$. Alternatively, we can also embed $\mathcal{P}_{XY}$ using the cross covariance operator $\mathcal{C}_{XY}$ [11], estimated by $D$ with:

$$\hat{\mathcal{C}}_{XY} = \frac{1}{s}\sum_{\ell=1}^{s}\phi_x(\mathbf{x}_\ell)\otimes\phi_y(y_\ell) = \frac{1}{s}\Phi_x^\top\Phi_y \in \mathbb{R}^{a\times b}.$$

where $\otimes$ denotes the outer product and similarly to $\mathcal{C}_{Y|X}$, we will flatten it to obtain $\psi(D) \in \mathbb{R}^{ab}$.

An important choice when modelling these quantities is the form of feature maps $\phi_x$, $\phi_y$ and $\phi_{xy}$, as these define the corresponding features of the data distribution we would like to capture. For example $\phi_x(\mathbf{x}) = \mathbf{x}$ and $\phi_x(\mathbf{x}) = \mathbf{x}\mathbf{x}^\top$ would be capturing the respective mean and second moment of the marginal distribution $P_x$. However, instead of defining a fixed feature map, here we will opt for a flexible representation, specifically in the form of neural networks (NN) for $\phi_x$, $\phi_y$ and $\phi_{xy}$ (except $\phi_y$ for classification[4]), in a similar fashion to [42]. To provide a better intuition on this choice, suppose we have two task $i, j$ and that $\mathcal{P}^i_{XY} \approx \mathcal{P}^j_{XY}$ (with the same sample size $s$). This will imply that $f^i \approx f^j$, and hence $\theta^*_i \approx \theta^*_j$. However, the converse does not hold in general: $f^i \approx f^j$ does *not* necessary imply $\mathcal{P}^i_{XY} \approx \mathcal{P}^j_{XY}$. For example, regularisation hyperparameters of a standard machine learning model are likely to be robust to rotations and orthogonal transformations of the covariates (leading to a different $P_X$). Hence, it is important to define a versatile model for $\psi(D)$, which can yield representations invariant to variations in the training data irrelevant for hyperparameter choice.

## 4.2 Modelling $f$

Given $\psi(D)$, we will now construct a model for $f(\theta, \mathcal{P}_{XY}, s)$, given observations $\left\{\{(\theta^i_k, \mathcal{P}^i_{XY}, s_i), z^i_k\}_{k=1}^{N_i}\right\}_{i=1}^n$, along with any observations on the target. Note that we will interchangeably use the notation $f$ to denote the model and the underlying function of interest. We will now focus on the algorithms distGP and distBLR, with additional details in Appendix A.

**Gaussian Processes (distGP)**. We proceed similarly to standard BO [34] using a GP to model $f$ and a normal likelihood (with variance $\sigma^2$ across all tasks[5]) for our observations $z$,

$$f \sim GP(\mu, C) \qquad z|\gamma \sim \mathcal{N}(f(\gamma), \sigma^2)$$

where here $\mu$ is a constant, $C$ is the corresponding covariance function on $(\theta, \mathcal{P}_{XY}, s)$ and $\gamma$ is a particular instance of an input. In order to fit a GP with inputs $(\theta, \mathcal{P}_{XY}, s)$, we use the following $C$:

$$C(\{\theta_1, \mathcal{P}^1_{XY}, s_1\}, \{\theta_2, \mathcal{P}^2_{XY}, s_2\}) = \nu k_\theta(\theta_1, \theta_2)k_p([\psi(D_1), s_1], [\psi(D_2), s_2])$$

where $\nu$ is a constant, $k_\theta$ and $k_p$ is the standard Matérn-$3/2$ kernel (with separate bandwidths across the dimensions). For classification, we additionally concatenate the class size ratio per class, as this

is not captured in $\psi(D_i)$. Utilising $\left\{ \{(\theta_k^i, \mathcal{P}_{XY}^i, s_i), z_k^i\}_{k=1}^{N_i} \right\}_{i=1}^n$, we can optimise $\mu$, $\nu$, $\sigma^2$ and any parameters in $\psi(D)$, $k_\theta$ and $k_p$ using the marginal likelihood of the GP (in an end-to-end fashion).

**Bayesian Linear Regression (distBLR)**. While GP with its well-calibrated uncertainties have shown superior performance in BO [34], it is well known that they suffer from $O(N^3)$ computational complexity [31], where $N$ is the total number of observations. In this case, as $N = \sum_{i=1}^n N_i$, we might find that the total number of evaluations across all tasks is too large for the GP inference to be tractable or that the computational burden of GPs outweighs the cost of computing $f$ in the first place. To overcome this problem, we will follow [26] and use Bayesian linear regression (BLR), which scales linearly in the number of observations, with the model given by

$$ z|\beta \sim \mathcal{N}(\Upsilon\beta, \sigma^2 I) \qquad \beta \sim \mathcal{N}(0, \alpha I) \qquad \Psi_i = [\psi(D_i), s_i] $$

$$ \Upsilon = [\upsilon([\theta_1^1, \Psi_1]), \ldots, \upsilon([\theta_{N_1}^1, \Psi_1]), \ldots, \upsilon([\theta_1^n, \Psi_n]), \ldots, \upsilon([\theta_{N_n}^n, \Psi_n])]^\top \in \mathbb{R}^{N \times d} $$

where $\alpha > 0$ denotes the prior regularisation, and $[\cdot, \cdot]$ denotes concatentation. Here $\upsilon$ denotes a feature map on concatenated hyperparameters $\theta$, data embedding $\psi(D)$ and sample size $s$. Following [26], we also employ a neural network for $\upsilon$. While conceptually similar to [26] who fits a BLR per task, here we consider a single BLR fitted jointly on all tasks, highlighting differences across tasks using meta-information available. The advantage of our approach is that for a given new task, we are able to utilise directly all previous information and one-shot predict hyperparameters without seeing *any* evaluations from the target task. This is especially important when our goal might be to employ our system with only a few evaluations from our target task. In addition, a separately trained target task BLR is likely to be poorly fitted given only a few evaluations. Similar to the GP case, we can optimise $\alpha, \beta, \sigma^2$ and any unknown parameters in $\psi(D), \upsilon([\theta, \Psi])$ using the marginal likelihood of the BLR.

## 4.3 Hyperparameter learning

Having constructed a model for $f$ and optimised any unknown parameters through the marginal likelihood, in order to construct a model for the $f^{\text{target}}$, we let $f^{\text{target}}(\theta) = f(\theta, \mathcal{P}_{XY}^{\text{target}}, s_{\text{target}})$. Now, to propose the next $\theta^{\text{target}}$ to evaluate, we can simply proceed with Bayesian optimisation on $f^{\text{target}}$, i.e. maximise the corresponding acquisition function $\alpha(\theta; f^{\text{target}})$. While we adopt standard BO techniques and acquisition functions here, note that the generality of the developed framework allows it to be readily combined with many advances in the BO literature, e.g. [12, 24, 19, 34, 41].

**Acquisition Functions**. For the form of the acquisition function $\alpha(\theta; f^{\text{target}})$, we will use the popular expected improvement (EI) [22]. However, for the first iteration, EI is not appropriate in our context, as these acquisition functions can favour $\theta$s with high uncertainty. Recalling that our goal is to quickly select 'good' hyperparameters $\theta$ with few evaluations, for the first iteration we will maximise the lower confidence bound (LCB)[6], as we want to penalise uncertainties and exploit our knowledge from source task's evaluations. While this approach works well for the GP case, for BLR, we will use the LCB restricted to the best hyperparameters from previous tasks, as BLR with a NN feature map does not extrapolate as well as GPs in the first iteration. For the exact forms of these acquisition functions, implementation and alternative warm-starting approaches, please refer to Appendix A.3.

**Optimisation**. We make use of ADAM [16] to maximise the marginal likelihood until convergence. To ensure relative comparisons, we standardised each task's dataset features to have mean $0$ and variance $1$ (except for the unsupervised toy example), with regression labels normalised individually to be in $[0, 1]$. As the sample size per task $s_i$ is likely to be large, instead of using the full set of samples $s_i$ to compute $\psi(D_i)$, we will use a different random sub-sample of batch-size $b$ for each iteration of optimisation (i.e. gradients are stochastic). In practice, this parameter $b$ depends on the number of tasks, and the evaluation cost of $f$. It should be noted that a smaller batch-size $b$ would still provide an unbiased estimate of $\psi(D_i)$ At testing time, it is also possible to use a sub-sample of the dataset to avoid any computational costs arising from a large $\sum_i s_i$. When retraining, we will initialise from the previous set of parameters, hence few gradient steps are required before convergence occurs.

**Extension to other data structures.** Throughout the paper, we focus on examples with $\mathbf{x} \in \mathbb{R}^p$. However our formulation is more general, as we only require the corresponding feature maps to be

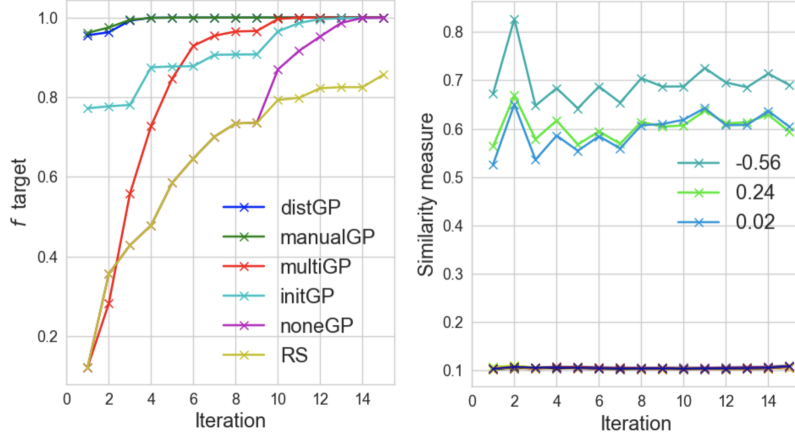

Figure 1: Unsupervised toy task over 30 runs. **Left**: Mean of the *maximum observed $f^{target}$* so far (including any initialisation). **Right:** Mean of the similarity measure $k_p(\psi(D_i), \psi(D_{\text{target}}))$ for distGP. For clarity purposes, the legend *only* shows the $\mu^i$ for the 3 source tasks that are similar to the target task with $\mu^i = -0.25$. It is noted the rest of the source task have $\mu^i \approx 4$.

defined on individual covariates and labels. For example, image data can be modelled by taking $\phi_x(\mathbf{x})$ to be a representation given by a convolutional neural network (CNN)[7], while for text data, we might construct features using Word2vec [21], and then retrain these representations for hyperparameter learning setting. More broadly, we can initialize $\psi(D)$ to any meaningful representation of the data, believed to be useful to the selection of $\theta^*_{\text{target}}$. Of course, we can also choose $\psi(D)$ simply as a selection of handcrafted meta-features [13, Ch. 2], in which case our methodology would use these representations to measure similarity between tasks, while performing feature selection [39]. In practice, learned feature maps via kernel mean embeddings can be used in conjunction with handcrafted meta-features, letting data speak for itself. In Appendix B.1, we provide a selection of 13 handcrafted meta-features that we employ as baselines for the experiments below.

## 5 Experiments

We will denote our methodology distBO, with BO being a placeholder for GP and BLR versions. For $\phi_x$ and $\phi_y$ we will use a single hidden layer NN with tanh activation (with 20 hidden and 10 output units), except for classification tasks, where we use a one-hot encoding for $\phi_y$. We further investigate this choice of NN structure in Appendix C.6 for the Protein dataset (results are fairly robust). For clarity purposes, we will focus on the approach where we separately embed the marginal and conditional distributions, before concatenation. Additional results for embedding the joint distribution can be found in Appendix C.1. For BLR, we will follow [26] and take feature map $\upsilon$ to be a NN with three 50-unit layers and tanh activation.

For baselines, we will consider: 1) manualBO with $\psi(D)$ as the selection of 13 handcrafted meta-features; 2) multiBO, i.e. multiGP [38] and multiBLR [26] where no meta-information is used, i.e. task is simply encoded by its index (they are initialised with 1 random iteration); 3) initBO [8] with plain Bayesian optimisation, but warm-started with the top 3 hyperparameters, from the three most similar source tasks, computing the similarity with the $\ell_2$ distance on handcrafted meta-features; 4) noneBO denoting the plain Bayesian optimisation [34], with no previous task information; 5) RS denoting the random search. In all cases, both GP and BLR versions are considered.

We use *TensorFlow* [1] for implementation, repeating each experiment 30 times, either through re-sampling (toy) or re-splitting the train/test partition (real life data). For testing, we use the same number of samples $s_i$ for toy data, while using a 60-40 train-test split for real data. We take the embedding batch-size[8] $b = 1000$, and learning rate for ADAM to be 0.005. To obtain $\{\theta^i_k, z^i_k\}^{N_i}_{k=1}$ for source task $i$, we use noneGP to simulate a realistic scenario. Additional details on these baselines

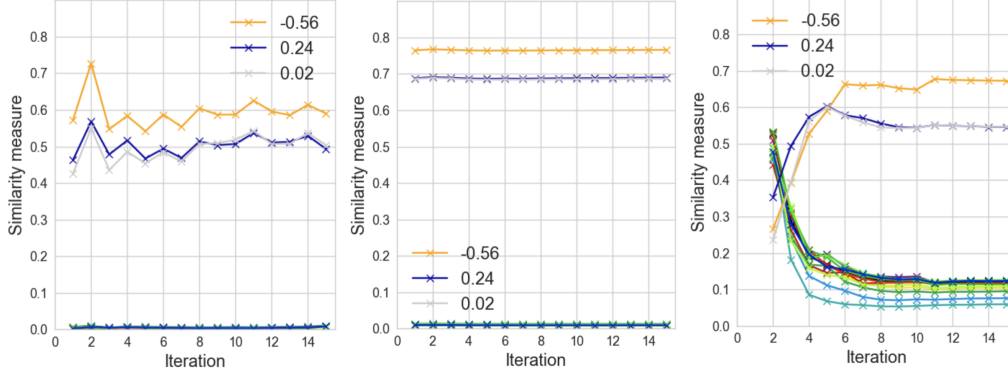

Figure 2: Mean of the similarity measure $k_p(\psi(D_i), \psi(D_{\text{target}}))$ over 30 runs versus number of iterations for the unsupervised toy task. For clarity purposes, the legend *only* shows the $\mu^i$ for the 3 source tasks that are similar to the target task with $\mu^i = -0.25$. It is noted the rest of the source task have $\mu^i \approx 4$. **Left:** distGP **Middle:** manualGP **Right:** multiGP

and implementation can be found in Appendix B and C, with additional toy (*non-similar source tasks scenario*) and real life (*Parkinson's dataset*) experiments to be found in Appendix C.4 and C.5.

## 5.1 Toy example.

To understand the various characteristics of the different methodologies, we first consider an "unsupervised" toy 1-dimensional example, where the dataset $D_i$ follows the generative process for some fixed $\gamma^i$: $\mu^i \sim \mathcal{N}(\gamma^i, 1)$; $x^i_\ell | \mu^i \overset{i.i.d.}{\sim} \mathcal{N}(\mu^i, 1)$. We can think of $\mu^i$ as the (unobserved) relevant property varying across tasks, and the unlabelled dataset as $D_i = \{x^i_\ell\}^{s_i}_{\ell=1}$. Here, we will consider the objective $f$ given by:

$$f(\theta; D_i) = \exp\left(-\frac{(\theta - \frac{1}{s_i}\sum_{\ell=1}^{s_i} x^i_\ell)^2}{2}\right),$$

where $\theta \in [-8, 8]$ plays the role of a 'hyperparameter' that we would like to select. Here, the optimal choice for task $i$ is $\theta = \frac{1}{s_i}\sum_{\ell=1}^{s_i} x^i_\ell$ and hence it is varying together with the underlying mean $\mu^i$ of the sampling distribution. An illustration of this experiment can be found in Figure 7 in Appendix C.2.

We now perform an experiment with $n = 15$, and $s_i = 500$, for all $i$, and generate 3 source tasks with $\gamma^i = 0$, and 12 source task with $\gamma^i = 4$. In addition, we generate an additional target dataset with $\gamma^{\text{target}} = 0$ and let the number of source evaluations per task be $N_i = 30$.

The results can be found in Figure 1. Here, we observe that distBO has correctly learnt to utilise the appropriate source tasks, and it is able to few-shot the optimum. This is also evident on the right of Figure 1, which shows the similarity measure $k_p(\psi(D_i), \psi(D_{\text{target}})) \in [0, 1]$ for distGP. The feature representation has correctly learned to place high similarity on the three source datasets sharing the same $\gamma^i$ and hence having similar values of $\mu^i$, while placing low similarity on the other source datasets. As expected, manualBO also few-shots the optimum here since the mean meta-feature which directly reveals the optimal hyperparameter was explicitly encoded in the hand-crafted ones. initBO starts reasonably well, but converges slowly, since the optimal hyperparameters even in the similar source tasks are not the same as that of the target task. It is also notable that multiBO is unable to few-shot the optimum, as it does not make use of any meta-information, hence needing initialisations from the target task to even begin learning the similarity across tasks. This is especially highlighted in Figure 2, which shows an incorrect similarity in the first few iterations. Significance is shown in the mean rank graph found in Figure 8 in Appendix C.2.

## 5.2 When handcrafted meta-features fail.

We now demonstrate an example in which using handcrafted meta-features does not capture any information about the optimal hyperparameters of the target task. Consider the following process for

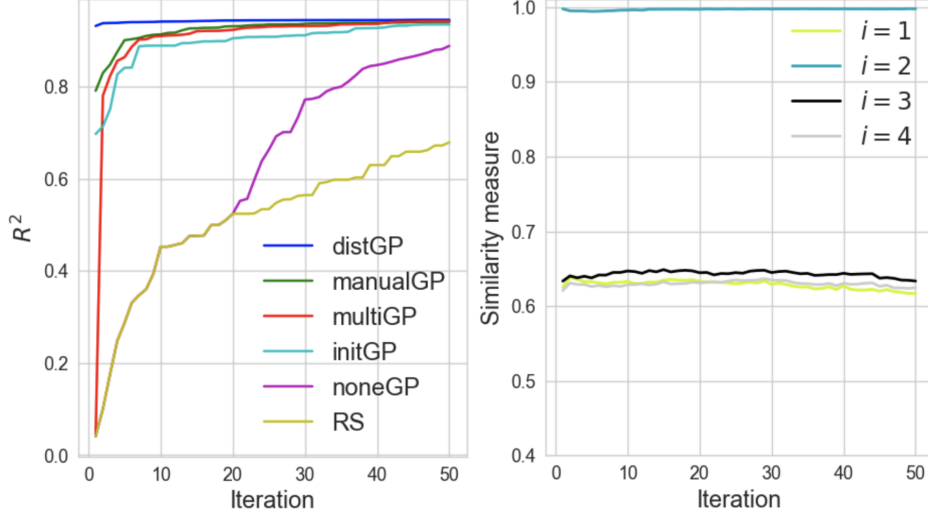

Figure 3: Handcrafted meta-features counterexample over 30 runs, with 50 iterations **Left**: Mean of the *maximum observed* $f^{target}$ so far (including any initialisation). **Right:** Mean of the similarity measure $k_p(\psi(D_i), \psi(D_{\text{target}}))$ for distGP, the target task uses the same generative process as $i = 2$.

dataset $i$ with $\mathbf{x}_\ell^i \in \mathbb{R}^6$ and $y_\ell^i \in \mathbb{R}$, given by:

$$
\begin{aligned}
\left[\mathbf{x}_\ell^i\right]_j &\overset{i.i.d.}{\sim} \mathcal{N}(0, 2^2), \quad j = 1, \ldots, 6, \\
\left[\mathbf{x}_\ell^i\right]_{i+2} &= \operatorname{sign}([\mathbf{x}_\ell^i]_1 [\mathbf{x}_\ell^i]_2) \left|[\mathbf{x}_\ell^i]_{i+2}\right|, \\
y_\ell^i &= \log\left(1 + \left(\prod_{j \in \{1, 2, i+2\}} [\mathbf{x}_\ell^i]_j\right)^3\right) + \epsilon_\ell^i.
\end{aligned}
\tag{1}
$$

where $\epsilon_\ell^i \overset{iid}{\sim} \mathcal{N}(0, 0.5^2)$, with index $i, \ell, j$ denoting task, sample and dimension, respectively: $i = 1, \ldots, 4$ and $\ell = 1, \ldots, s_i$ with sample size $s_i = 5000$. Thus across $n = 4$ source tasks, we have constructed regression problems, where the dimensions which are relevant (namely 1, 2 and $i + 2$) are varying. Note that (1) introduces a three-variable interaction in the relevant dimensions, but that all dimensions remain pairwise independent and identically distributed. Thus, while these tasks are inherently different, this difference is invisible by considering marginal distribution of covariates and their pairwise relationships such as covariances. As the handcrafted meta-features for manualBO only consider statistics which process one or two dimensions at the time or landmarkers [27], their corresponding $\psi(D_i)$ are *invariant* to tasks up to sampling variations. For an in-depth discussion, see Appendix C.3. We now generate an additional target dataset, using the same generative process as $i = 2$, and let $f$ be the coefficient of determinant ($R^2$) on the test set resulting from an automatic relevance determination (ARD) kernel ridge regression with hyperparameters $\alpha$ and $\sigma_1, \ldots, \sigma_6$. Here $\alpha$ denotes the regularisation parameter, while $\sigma_j$ denotes the kernel bandwidth for dimension $j$.

Setting $N_i = 125$, the results can be found in Figure 3 (GP) and Figure 9 in Appendix C.3 (BLR). It is clear that while distBO is able to learn a high similarity to the correct source task (as shown in Figure 3), and one-shot the optimum, this is not the case for any of the other baselines (Figure 10 in Appendix C.3) . In fact, as manualBO's meta-features do not include any useful meta-information, they essentially encode the task index, and hence perform similarly to multiBO. Further, we observe that initBO has slow convergence after warm-starting. This is not surprising as initBO has to 're-explore' the hyperparameter space as it only uses a subset of previous evaluations. This highlights the importance of using all evaluations from all source tasks, even if they are sub-optimal. In Figure 9 in Appendix C.3, we show significance using a mean rank graph and that the BLR methods performs similarly to their GP counterparts.

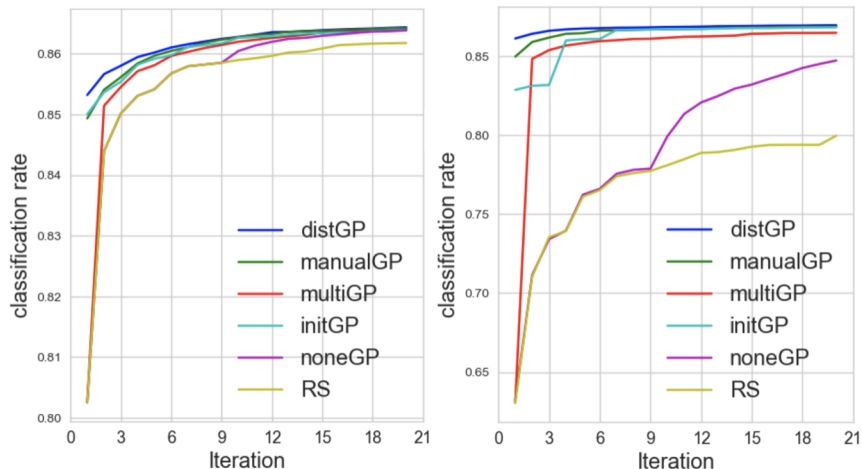

Figure 4: Each evaluation is the *maximum observed* accuracy rate averaged over 140 runs, with 20 runs on each of the protein as target. **Left:** Jaccard kernel C-SVM. **Right:** Random forest

### 5.3 Classification: Protein dataset.

We now apply the methodologies to a real life protein-ligand binding problem in the area of drug design. In particular, the Protein dataset consists of 7 different proteins extracted from [9]: ADAM17, AKT1, BRAF, COX1, FXA, GR, VEGFR2. Each protein dataset contains $1037 - 4434$ molecules (data-points $s_i$), where each molecule has binary features $\mathbf{x}_\ell^i \in \mathbb{R}^{166}$ computed using a chemical fingerprint (MACCs Keys[9]). The label per molecule is whether the molecule can bind to the protein target $\in \{0, 1\}$. In this experiment, we can treat each protein as a separate classification task. We consider two classification methods: Jaccard kernel C-SVM [5, 30] (commonly used for binary data, with hyperparameter $C$), and random forest (with hyperparameters $n\_trees$, $max\_depth$, $min\_samples\_split$, $min\_samples\_leaf$), with the corresponding objective $f$ given by accuracy rate on the test set. In this experiment, we will designate each protein as the target task, while using the other $n = 6$ proteins as source tasks. In particular, we will take $N_i = 20$ and hence $N = 120$. The results obtained by averaging over different proteins as the target task (20 runs per task) are shown in Figure 4 (with mean rank graphs and BLR version to be found in Figure 14 and 15 in Appendix C.6). On this dataset, we observe that distGP outperforms its counterpart baselines and few-shots the optimum for both algorithms. In addition, we can see a slower convergence for the multiGP and initGP, demonstrating the usefulness of meta information in this context.

## 6 Conclusion

We demonstrated that it is possible to borrow strength between multiple hyperparameter learning tasks by making use of the similarity between training datasets used in those tasks. This helped us to develop a method which finds a favourable setting of hyperparameters in only a few evaluations of the target objective. We argue that the model performance should not be treated as a black box function as it corresponds to specific known models and specific datasets. We demonstrate that its careful consideration as a function of all its inputs, and not just of its hyperparameters, can lead to useful algorithms.

## 7 Acknowledgements

We thank Kaspar Martens, Jin Xu, Wittawat Jitkrittum and Jean-Francois Ton for useful discussions. HCLL is supported by the EPSRC and MRC through the OxWaSP CDT programme (EP/L016710/1). DS is supported in part by the ERC (FP7/617071) and by The Alan Turing Institute (EP/N510129/1). HCLL partially completed this work at Tencent AI Lab, and HCLL and DS are supported in part by the Oxford-Tencent Collaboration on Large Scale Machine Learning.

## Footnotes

[2]A comprehensive survey on meta-learning and handcrafted meta-features can be found in [13, Ch.2], [8]

[3]Following [17], in practice we re-scale $s$ to $[0, 1]$, so that the task with the largest sample size has $s = 1$.

[4]For classification, we use $\hat{\mathcal{C}}_{XY}$ and a one-hot encoding for $\phi_y$ implying a marginal embedding per class.

[5]For different noise levels across tasks, we can allow for different $\sigma^2_i$ per task $i$ in distGP and distBLR.

[6]Note this is not the upper confidence bound, as we want to *exploit* and obtain a good starting initialisation.

[7]This is similar to [18] who embeds distribution of images using a pre-trained CNN for distribution regression.

[8]Training time is less than 2 minutes on a standard 2.60GHz single-core CPU in all experiments.

[9]http://rdkit.org/docs/source/rdkit.Chem.MACCSkeys.html

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
