[Supplementary Material · bo_appendix.pdf]

# A    Additional details for methodology

## A.1    Gaussian process (distGP)

For distGP, we have the following model:

$$
\begin{aligned}
f &\sim GP(\mu, C) \\
z|\gamma &\overset{i.i.d.}{\sim} \mathcal{N}(f(\gamma), \sigma^2)
\end{aligned}
$$

where here $\mu$ is taken to be a constant and $C$ is the corresponding covariance function. In this case, the log marginal likelihood with observations $\Gamma = \left\{ \{(\theta_k^i, \mathcal{P}_{XY}^i, s_i), z_k^i\}_{k=1}^{N_i} \right\}_{i=1}^{n}$, following standard GP literature [31] is given by:

$$
\log(p(\mathbf{z}|\Gamma)) = -\frac{1}{2}(\mathbf{z} - \mu)^\top (K + \sigma^2 I)^{-1}(\mathbf{z} - \mu) - \frac{1}{2}\log|K + \sigma^2 I| - \frac{N}{2}\log(2\pi)
$$

where $\mathbf{z} = [z_1^1, \ldots z_{N_n}^n]^\top$, $N = \sum_i N_i$ and $K$ is the kernel matrix, with $K_{ij} = C(\gamma_i, \gamma_j)$. Here $\gamma_i, \gamma_j$ denotes elements of $\Gamma$. In particular, for a new observation $\gamma^*$, the predictive posterior distribution $f_{\text{post}}(\gamma^*) \sim \mathcal{N}(\mu_{\text{post}}(\gamma^*), \sigma_{\text{post}}^2(\gamma^*))$, where:

$$
\begin{aligned}
\mu_{\text{post}}(\gamma^*) &= \mu + K_{\gamma^*\Gamma}(K + \sigma^2 I)^{-1}(\mathbf{z} - \mu) \\
\sigma_{\text{post}}^2(\gamma^*) &= K_{\gamma^*\gamma^*} - K_{\gamma^*\Gamma}(K + \sigma^2 I)^{-1}K_{\gamma^*\Gamma}^\top
\end{aligned}
$$

where here $K_{\gamma^*\gamma^*} = C(\gamma^*, \gamma^*)$ and $K_{\gamma^*\Gamma} = [C(\gamma^*, \gamma_1), \ldots, C(\gamma^*, \gamma_N)]$.

## A.2    Bayesian Linear Regression (distBLR)

$$
z|\beta \overset{i.i.d.}{\sim} \mathcal{N}(\Upsilon\beta, \sigma^2 I) \qquad \beta \sim \mathcal{N}(0, \alpha I)
$$

where $\Upsilon = [\upsilon([\theta_1^1, \psi(D_1), s_1]), \ldots, \upsilon([\theta_{N_n}^n, \psi(D_n), s_n])]^\top \in \mathbb{R}^{N \times d}$ and $\alpha > 0$ denotes the prior regularisation. Here $\upsilon$ denotes a feature map of dimension $d$ on concatenated hyperparameters $\theta$, data embedding $\psi(D)$ and sample size $s$. Following [3, 26], defining $K_{\text{dim}} = I_d + \frac{\alpha}{\sigma^2}\Upsilon^\top\Upsilon$, and $L$ as the cholesky factor of $K_{\text{dim}}$, i.e. $K_{\text{dim}} = LL^\top$, the log marginal likelihood (up to additive constants) with observations $\Gamma = \left\{ \{(\theta_k^i, \mathcal{P}_{XY}^i, s_i), z_k^i\}_{k=1}^{N_i} \right\}_{i=1}^{n}$ is given by:

$$
\log(p(\mathbf{z}|\Gamma)) = \frac{1}{2\sigma^2}\left(\frac{\alpha}{\sigma^2}||\mathbf{e}||^2 - ||\mathbf{z}||^2\right) - \sum_{i=1}^{d}\log(L_{ii}) - \frac{N}{2}\log(\sigma^2)
$$

where $\mathbf{e} = L^{-1}\Upsilon^\top\mathbf{z}$. In this case, for a given $\boldsymbol{v}^* \in \mathbb{R}^{d \times 1}$, the transformed feature map of a particular instance of $\gamma^*$, the predictive posterior distribution $\beta^\top\boldsymbol{v}^* = f_{\text{post}}(\gamma^*) \sim \mathcal{N}(\mu_{\text{post}}(\gamma^*), \sigma_{\text{post}}^2(\gamma^*))$, where:

$$
\begin{aligned}
\mu_{\text{post}}(\gamma^*) &= \frac{\alpha}{\sigma^2}\mathbf{e}^\top L^{-1}\boldsymbol{v}^* \\
\sigma_{\text{post}}^2(\gamma^*) &= \alpha||L^{-1}\boldsymbol{v}^*||^2
\end{aligned}
$$

It is noted that the computational complexity here scales linearly in the number of observations $N$ and cubically in $d$.

## A.3    Warm-starting, acquisition functions and multi-task extension

The lower confidence bound (LCB) [37] is defined as follows:

$$
\alpha_{\text{LCB}}(\gamma; f_{\text{post}}) = \mu_{\text{post}}(\gamma) - \kappa * \sigma_{\text{post}}(\gamma)
$$

where $\kappa$ denotes the level of exploration, and for experiments we set $\kappa = 2.58$, as we would like to exploit the information from other tasks on our first iteration. It should be noted that this is not the upper confidence bound commonly used, as we would like to penalise uncertainty on the first iteration.

The expected improvement (EI) [22] is defined as follows:

$$
\begin{aligned}
g(\gamma) &= (\mu_{\text{post}}(\gamma) - z_{\text{max}} - \xi)/\sigma_{\text{post}}(\gamma) \\
\alpha_{\text{EI}}(\gamma; f_{\text{post}}) &= \sigma_{\text{post}}(\gamma)(g(\gamma)\Phi_{\text{cdf}}(g(\gamma)) + \mathcal{N}(g(\gamma); 0, 1)
\end{aligned}
$$

where here $z_{\max}$ refers to the maximum observed $z$ for our *target task*, while $\Phi_{\text{cdf}}$ and $\mathcal{N}(g(\gamma); 0, 1)$ refers to the CDF and pdf of a standard Normal distribution. For experiments, we set the exploration parameter to be $\xi = 0.01$. It should be noted in the case, where the $\alpha_{\text{EI}} = 0$ (or numerically close to 0) for all attempted locations, we will use the upper confidence bound (with $\kappa = 2.58$) [37] instead. To maximise the acquisition function, we first randomly select $300,000$ hyperparameters for evaluation (computationally cheap), to find the top 10 optimum. Initialising from these top 10 hyperparameters, a L-BFGS-B algorithm (computationally expensive) is used to maximise the acquisition function, to select the next hyperparameter for evaluation.

**Warm-starting** Instead of using the LCB acquisition function (for the first evaluation), an alternative approach is to warm-start [10, 32, 8] based on *learnt* similarities with previous source tasks. For the GP case, we will optimise the marginal likelihood based on all observations from the source tasks, learning the task similarity function $k_p([\psi(D_i), s_i], [\psi(D_j), s_j])$. As the output domain of $k_p$ lies in $[0, 1]$, we can compute the top $M$ source tasks most similar with our target task. Given this selection, we can extract the best $m$ previous best hyperparameters from each of these source tasks, enabling $Mm$ hyperparameters as warm-start initialisations for our algorithm. For the BLR case, as a joint space over $\theta$, $\psi(D)$ and $s$ is considered, a direct task similarity function is no longer available. Instead we opt for a different approach and extract $m$ previous best hyperparameters from all source tasks, and consider only these hyperparameters for the maximisation of the LCB/EI acquisition function. In practice, we recommend to warm-start with as few evaluations as possible, as:

- Source tasks can be dissimilar to our target task.
- Warm-start hyperparameters may be similar to each other, and hence costly evaluations are either wasted or inefficient.
- More evaluations are needed before the proposed algorithm can begin to utilise all seen evaluations to explore/exploit for our target task.

# B    Baselines

## B.1    manualBO

Instead of constructing $\psi(D)$, as described in section 4, we can select $\psi(D)$ to be a selection of handcrafted meta-features. Here, we provide the set of meta-features we used for experiments. It should be noted that features of $X^i = \{\mathbf{x}^i_\ell\}^{s_i}_{\ell=1}$ is standardised to have mean 0 and variance 1 individually (except for the unsupervised toy example case, in which we encode the mean meta-feature explicitly), while $y^i_\ell$ is normalised to be in $[0, 1]$ for regression. To ensure fair relative comparisons, meta-features are normalised to be in $[0, 1]$ across all tasks [2]. We do not include sample size $s_i$, as these are already encoded separately.

**General meta-features**

- Skewness, kurtosis [20]: these are calculated on each feature of the dataset $X^i$, before the minimum, maximum, mean and standard deviation of the computed quantities is extracted across the features.
- Correlation, covariance [20]: these are calculated on every pair of features of $X^i$, before the minimum, maximum, mean and standard deviation of the computed quantities is extracted across each pair of features.
- PCA skewness, kurtosis [7]: principal component analysis (PCA) is performed on $X^i$, and $X^i$ is projected onto the first principal component. The corresponding skewness and kurtosis is computed.
- Intrinsic dimensionality [2]: number of principal components to explain $95\%$ of variance.

**Classification specific meta-features**

- Class ratios, entropy [20]: empirical class distribution and its corresponding entropy.
- Classification landmarkers [27]: 1-nearest-neighbour classifier, linear discriminant analysis, naive Bayes and decision tree classifier.

**Regression specific meta-features**

- Mean, standard deviation, skewness, kurtosis of the labels $\{y_\ell^i\}_{\ell=1}^{s_i}$ [20].
- Regression landmarkers [27]: 1-nearest-neighbour regressor, linear regression and decision tree regressor.

The landmarkers are scalable algorithms that are cheap to run, and provide us various characteristic of the machine learning task. The corresponding meta-feature from these landmarkers is the accuracy on an independent set of data (a train-test split is done on $X^i$, the training data). In experiments, we use the default settings in *sklearn* [25] for these algorithms. For additional details on their formulation and rationale, please refer to [13, Ch.2].

## B.2  multiBO

Instead of using meta-features, we may wish to simply encode the task index, and learn task similarities based on only $\left\{ \{\theta_k^i, z_k^i\}_{k=1}^{N_i} \right\}_{i=1}^{n}$. It should be noted that in both these cases, we do not encode any sample size or class ratio information and initial evaluations from the target task is required.

**multiGP**  For the GP case, we will follow [38], who considers a multi-task GP for Bayesian optimisation. Instead of using the kernel $k_p$ on meta-features, we will now replace it by a kernel on tasks $k_t$. Given the $n + 1$ total number of tasks (including the target task), the task similarity matrix is given by $S_t = L_t L_t^T \in \mathbb{R}^{n+1 \times n+1}$, where $L_t$ is a learnt cholesky factor. Expanding $S_t$ into the appropriate sized kernel $K_t \in \mathbb{R}^{N \times N}$ (as we have repeated observations from the same task), using the marginal likelihood, we can learn the lower triangular elements of $L_t$. Similar to [38], we assume positive correlation amongst tasks and restrict positivity in the elements of the cholesky factor.

**multiBLR**  For the BLR case, we will follow [26] and consider a one-hot encoding for $\psi(D_i)$. This representation essentially identifies a separate encoding for every task, and similarity between tasks (and hyperparameters) is captured through the transformation $\upsilon$ (without sample size $s_i$), which we learn using the marginal likelihood.

## B.3  initBO

For this baseline, we will employ the handcrafted meta-features as described in Appendix B.1 to warm-start Bayesian optimisation, using a GP or BLR. In particular, we first define the number of evaluations $m$ per task and the number of tasks $M$ we wish to warm-start with (i.e. $Mm$ number of warm-start hyperparameters). To define a similarity function, for a fair comparison with existing literature, we will use the $\ell_2$ norm [8] between the datasets' meta-features:

$$k(D_i, D_j) = -|| \, [\psi(D_i), s_i] - [\psi(D_j), s_j] \, ||_2$$

where here $k$ is a similarity function, and $\psi(D_i)$ is the handcrafted meta-features representation for task $i$. It should also be noted that as meta-features are individually normalised to be in $[0, 1]$, no particular meta-feature is emphasised in this distance measure. To obtain the warm-start $\theta$s, we compute $k(D_{\text{target}}, D_j)$ for all $j = 1, \ldots, n$ and extract the $M$ tasks with highest similarity. Given these $M$ tasks, we extract the $m$ best performing hyperparameters from each of these task to obtain $Mm$ warm-start hyperparameters. These hyperparameters will then be used for warm-starting noneGP or noneBLR (instead of random evaluations).

## C  Experiments

With the exception of the hyperparameter in the unsupervised toy and the protein random forest example, all other hyperparameters are optimised in the log-scale. In addition, we standardise hyperparameters to have mean 0 and variance 1, when passing them to the GP and BLR, to ensure parameters initialisation are well-defined. Here we provide additional details for our experiments in section 5.

## C.1 Comparison between joint and concatenation embeddings for regression

Here we display additional graphs comparing the embedding of the joint distribution versus the embedding of the conditional distribution and marginal distribution before concatenation. We denote these correspondingly by distGP-joint, distBLR-joint and distGP-concat, distGP-concat. Overall, we observe that their performance is similar.

Figure 5: Manual meta-features counterexample with 50 iterations (including any initialisation). Here, BLR methods are displayed on the top, while GP methods are displayed on the bottom. Each evaluation here is averaged over 30 runs. **Left:** *Maximum observed $R^2$*. **Right:** Mean rank (with respect to each run) of the different methodologies, with $\pm 1$ sample standard deviation.

Figure 6: Parkinson's experiment with 17 iterations (including any initialisation). Each evaluation here is averaged over $420$ runs, with each of the $42$ patient set as the target task (repeated for $10$ runs) **Left:** *Maximum observed $R^2$*. **Right:** Mean rank (with respect to each run) of the different methodologies, with $\pm 1$ sample standard deviation.

## C.2 Unsupervised toy example

Hyperparameters: $\theta \in [-8, 8]$
Source task's random and BO iterations: $10, 20$
Target task's noneBO random and BO iterations: $5, 10$
An illustration of this toy example can be seen in figure 7.

Figure 7: Illustration of unsupervised toy example.

Figure 8: Unsupervised toy task with 15 iterations (including any initialisation). Each evaluation here is averaged over 30 runs. **Left:** *Maximum observed* $f^{target}$. **Right:** Mean rank (with respect to each run) of the different methodologies, with $\pm 1$ sample standard deviation.

## C.3 Regression: handcrafted meta-features counterexample

Hyperparameters: $\alpha \in [10.0^{-8}, 0.1], \sigma_j \in [2.0^{-7}, 2.0^5]$
Source task's random and BO iterations: $50, 75$
Target task's noneBO random and BO iterations: $20, 30$

For task $i = 1, \ldots 4$, we have the process:

$$
\begin{aligned}
\left[\mathbf{x}_\ell^i\right]_j &\sim \mathcal{N}(0, 2^2) \quad j = 1, \ldots, 6 \\
\left[\mathbf{x}_\ell^i\right]_{i+2} &= \operatorname{sign}([\mathbf{x}_\ell^i]_1 [\mathbf{x}_\ell^i]_2) \left|[\mathbf{x}_\ell^i]_{i+2}\right| \\
y_\ell^i &= \log\left(1 + \left(\prod_{j \in \{1,2,i+2\}} [\mathbf{x}_\ell^i]_j\right)^3\right) + \epsilon_\ell^i
\end{aligned}
$$

where $\epsilon_\ell^i \overset{iid}{\sim} \mathcal{N}(0, 0.5^2)$, with index $i, \ell, j$ denoting task, sample and dimension. For each task $i$, the dimension of importance is $1, 2$ and $i + 2$, while the rest is nuisance variables. We now demonstrate that the handcrafted meta-features for regression in Appendix B.1 do not differ across the tasks (when noise is not considered). Firstly, it is noted that $\left[\mathbf{x}_\ell^i\right]_{i+2} \sim \mathcal{N}(0, 2^2)$ even after alteration. This then implies that meta-features measuring skewness and kurtosis per dimension does not change across tasks. Similarly, any PCA meta-features will remain the same, as variances remains the same in all directions. Further, as $\left[\mathbf{x}_\ell^i\right]_{i+2}$ remains independent to $\left[\mathbf{x}_\ell^i\right]_j$ for $j \neq k$, meta-features based on correlation and covariance will remain to be $0$ for all pairs of features. Lastly, for regression landmarkers and labels, as these are not perturbed by permutation of the features of the dataset, the regression specific meta-features also remains the same. Together, this implies that the handcraft meta-features are unable to distinguish which source task is similar to the target task (with the same process as $i = 2$). However, as we have additional noise samples for each task, the computed representation $\psi(D_i)$ still differs amongst all the tasks, hence the specific task can still be recognised.

Figure 9: Manual meta-features counterexample with 50 iterations (including any initialisation). Here, GP methods are displayed on the left, while BLR methods are displayed on the right. Each evaluation here is averaged over 30 runs. **Top row:** *Maximum observed $R^2$*. **Bottom row:** Mean rank (with respect to each run) of the different methodologies, with $\pm 1$ sample standard deviation.

Figure 10: Mean of the similarity measure $k_p(\psi(D_i), \psi(D_{\text{target}}))$ over 30 runs versus number of iterations for the manuak meta-features counterexample. The target task uses the same generative process as $i = 2$. **Left:** distGP **Middle:** manualGP **Right:** multiGP

## C.4 Classification: similar and not similar source tasks

Hyperparameters: $C \in [2.0^{-7}, 2.0^{10}], \sigma_j \in [2.0^{-3}, 2.0^5]$
Source task's random and BO iterations: $75, 75$
Target task's noneBO random and BO iterations: $25, 75$

Figure 11: Classification task experiment A with 100 iterations (including any initialisation). Here, the target task is similar to one of the source task. Each evaluation here is averaged over 30 runs. **Left:** *Maximum observed* AUC. **Right:** Mean rank (with respect to each run) of the different methodologies, with $\pm 1$ sample standard deviation.

Figure 12: Classification task experiment B with 100 iterations (including any initialisation). Here the target task is *different* to all the source task. Each evaluation here is averaged over 30 runs. **Left:** *Maximum observed* AUC. **Right:** Mean rank (with respect to each run) of the different methodologies, with $\pm 1$ sample standard deviation.

We now demonstrate a classification example, where we contrast the case where some of the source tasks is similar to the target tasks against the case where no such source task exists to illustrate that encoding meta-information need not always be beneficial. Here, we let the number of source

tasks $n = 10$, $s_i = 5000$ and $f$ to be the AUC on the test set for ARD kernel logistic regression, with hyperparameters $C$ and $\sigma_1, \ldots, \sigma_6$. Similar to before, $C$ denotes regularisation and $\sigma_j$ denotes the kernel bandwidth for dimension $j$. To generate $D_i$, we take $\mathbf{x}_\ell^i \sim \mathcal{N}(\mathbf{0}, I_6)$, and obtain $y_\ell^i$ conditionally on $\mathbf{x}_\ell^i$ by sampling from a kernel logistic regression model (ARD kernel with Random Fourier features [29] approximation) where each task has different "true" bandwidth parameters (also different across dimensions).

To be more precise, to generate $\{\mathbf{x}_\ell^i, y_\ell^i\}_{\ell=1}^{s_i}$ for this experiment, we first simulate $\mathbf{x}_\ell^i \sim \mathcal{N}(\mathbf{0}, I_6)$. Then in order to sample from the model of an ARD kernel logistic regression, we define an underlying true bandwidth $\tilde{\boldsymbol{\sigma}}^i = [\tilde{\sigma}_1^i, \ldots, \tilde{\sigma}_6^i]$ and use random Fourier features (RFF) [29] to approximate an ARD kernel (with $D = 200$ frequencies) as follows:

$$\boldsymbol{\varphi}_\ell^i = \sqrt{2/D}\cos(\mathbf{U}\tilde{\mathbf{x}}_\ell^i + \mathbf{b}) \qquad \mathbf{U} \in \mathbb{R}^{D\times 6}, \mathbf{b} \in \mathbb{R}^D$$

where $\tilde{\mathbf{x}}_\ell^i = \mathbf{x}_\ell^i/\tilde{\boldsymbol{\sigma}}^i$ denotes element-wise division by the bandwidths in respective dimensions and $\mathbf{U}_{mn} \overset{i.i.d.}{\sim} \mathcal{N}(0,1)$ and $\mathbf{b}_m \overset{i.i.d.}{\sim} \text{Unif}([0, 2\pi])$. Letting $\boldsymbol{\Phi}^i = [\boldsymbol{\varphi}_1^i, \ldots \boldsymbol{\varphi}_{s_i}^i]^\top$, we let $\tilde{\mathbf{g}}^i = \boldsymbol{\Phi}^i \boldsymbol{\beta}^i$, where $\boldsymbol{\beta}^i \sim \mathcal{N}(0, I_D)$. We then normalise $\tilde{\mathbf{g}}^i$ to be in the range $[-6, 6]$ and then transform it through the logistic link:

$$p_\ell^i = \frac{1}{1 + \exp(-\tilde{g}_\ell^i)}$$

obtaining $p_\ell^i = P(y_\ell^i = 1|x_\ell^i)$, using which we can draw a binary output $y_\ell^i \sim \text{Bernoulli}(p_\ell^i)$. For the source tasks, we will randomly select $\tilde{\sigma}_j^i \in \{0.5, 1.0, 2.0, 4.0, 8.0, 16.0\}$ with replacement across all $j$, so that different dimensions are of different relative importance across different tasks. For experiment A, we will select its underlying bandwidths to be the same as one of that in the source task. For experiment B, to ensure that our target task has different optimal hyperparameters to the source tasks, we will let $\tilde{\sigma}_j^i = 1.5$ for all $j$.

Note that all tasks have the same marginal distribution of covariates and that there is a high variation in conditional distributions: they differ not only in terms of kernel bandwidths but also in terms of coefficients in their respective regression functions. To generate a task dataset, we use the same process, and run 2 experiments: (A) use the same set of bandwidths as one of the source tasks but a different regression function, and (B) use a set of bandwidths unseen in any of the source tasks (and a different regression function).

We take $N_i = 150$ and since the total number of evaluations is $N = 1500$, we focus our attention on BLR, which have $O(N)$ linear computational complexity. The results for the two experiments are shown in Figure 11 and 12. We see that distBLR leverages the presence of a similar task among the sources and learns a representation of the dataset which helps guide hyperparameter selection to the optimum faster than other methods. We note that manualBLR converges much slower, given that the optimal hyperparameters depend on the data in a complex way which is difficult to extract from handcrafted meta-features. We also note that initBLR performs poorly despite the presence of a source task with the *same "true" bandwidths*: often, the meta-features are not powerful enough to recognize which task is the most similar in order to initialise appropriately.

On the other hand, in the case B, no similar source exists implying that the joint BLR model in distBLR needs to extrapolate to the far away region in the space of joint distributions of training data. As expected, meta-information in this example is not as helpful as in the case A and the method that ignores it, multiBLR, in fact performs best. However, albeit worse performing, note that distBLR and manualBLR were still able to revert to the behaviour akin to multiBLR and achieve a faster convergence compared to their non-transfer counterparts and initBLR which essentially has to re-explore the hyperparameter space from scratch.

## C.5   Regression: Parkinson's dataset

Hyperparameters: $\alpha \in [10.0^{-10}, 0.1], \sigma_j \in [2.0^{-7}, 2.0^5]$
Source task's random and BO iterations: $10, 20$
Target task's noneBO random and BO iterations: $9, 8$

Figure 13: Parkinson's experiment with 17 iterations (including any initialisation). Each evaluation here is averaged over 420 runs, with each of the 42 patient set as the target task (repeated for 10 runs) **Left:** *Maximum observed* $R^2$. **Right:** Mean rank (with respect to each run) of the different methodologies, with $\pm 1$ sample standard deviation.

The Parkinson's disease telemonitoring dataset[10] consists of voice measurements using a telemonitoring device for 42 patients with Parkinson disease (approximately $150$ recordings $\in \mathbb{R}^{17}$ each). The label is the clinician's Parkinson disease symptom score for *each recording*. Following a setup similar to [4], we can treat each patient as a separate regression task. In this experiment, in order to allow for comprehensive benchmark comparisons, we consider $f$ which is not prohibitively expensive (hence the problem does not necessarily benefit computationally from Bayesian optimisation). Namely, we employ RBF kernel ridge regression (with hyperparameters $\alpha, \gamma$), with $f$ as the coefficient of determination ($R^2$). In this experiment, we will designate each patient as the target task, while using the other $n = 41$ patients as source tasks. In particular, we will take $N_i = 30$, and hence $N = 1230$, and again since the total number of evaluations is large, will focus on BLR. The results obtained by averaging over different patients as the target task (20 runs per task) are shown in Figure 13. On this dataset, we observe similar behaviour of transfer methods which were able to leverage the source task information and for many patients few-shot the optimum. This suggests the presence of similar source tasks in practice and that this similarity can be exploited in the context of hyperparameter learning.

## C.6 Classification: protein dataset

**Jaccard kernel C-SVM**
Hyperparameters: $C \in [2.0^{-7}, 2.0^{10}]$
Source task's random and BO iterations: $10, 10$
Target task's noneBO random and BO iterations: $9, 11$

To compute the Jaccard kernel [5, 30], we use of the python package *SciPy*[11] [14] to compute the Jaccard distance, before performing a one subtract each entry to get a similarity matrix. Results are shown in Figure 14.

Figure 14: Protein dataset with Jaccard kernel C-SVM. Each evaluation here is averaged over 140 runs, with each of the 7 protein set as the target task (20 runs each). GP methods are displayed on the left, while BLR methods are displayed on the right. **Top row:** *Maximum observed* classification accuracy $(\%)$. **Bottom row:** Mean rank (with respect to each run) of the different methodologies, with $\pm 1$ sample standard deviation.

**Random Forest**

Hyperparameters:

Number of trees: $n\_trees \in \{1, \ldots, 200\}$

Max depth of the tree: $max\_depth \in \{1, \ldots, 32\}$

Min samples required to split a node (after multiplied with $s_i$): $min\_samples\_split \in [0.01, 1.0]$

Min samples required at a leaf node (after multiplied with $s_i$): $min\_samples\_leaf \in [0.01, 0.5]$

Source task's random and BO iterations: $10, 10$

Target task's noneBO random and BO iterations: $9, 11$

Since $n\_trees$ and $max\_depth$ are discrete hyperparameters, in practice we round up to the nearest integer, after a continuous version of it is proposed. For additional information on these hyperparameters, please refer to the *RandomForestClassifier*[12] in the Python package *scikit-learn* [25]. Results are shown in Figure 15.

Figure 15: Protein dataset with random forest. Each evaluation here is averaged over 140 runs, with each of the 7 protein set as the target task (20 runs each). GP methods are displayed on the left, while BLR methods are displayed on the right. **Top row:** *Maximum observed* classification accuracy (%). **Bottom row:** Mean rank (with respect to each run) of the different methodologies, with $\pm 1$ sample standard deviation.

**Varying NN structure**

We choose a small two layer neural network (20 hidden and 10 output units) to avoid over-parameterisation, as the number of evaluations, $N$ is small. Here, we also provide experiments in Figure 16 to show that our methodology is indeed robust to this choice. It is noted that here for classification, we use a one hot encoding for $\phi_y$, and that here the difference of accuracy between various NN structures is on order of $0.2\% - 0.3\%$ only.

Figure 16: Protein dataset with random forest across 140 evaluations with different NN structure for distGP's $\phi_x$. The legend here represents the precise neural network structure, e.g. 20-10 implies 20 hidden and 10 output units.

## Footnotes

[10]http://archive.ics.uci.edu/ml/datasets/Parkinsons+Telemonitoring

[11]https://docs.scipy.org/doc/scipy/reference/generated/scipy.spatial.distance.cdist.html

[12]https://scikit-learn.org/stable/modules/generated/sklearn.ensemble.RandomForestClassifier.html