[Reviews · NeurIPS 2019]

Reviewer 1



[I have read the author rebuttal. Since it does address my essential concern of whether the performance observed in the paper generalizes, I've upgraded my score.] Summary: I liked the step in the direction of a more principled and generic approach for leveraging dataset characteristics for hyperparameter selection. However, the limitation to a single model class and instance space appear to constrain broad applicability. Moreover, even in this restricted setting, there is a fairly large design space to consider: embedding distribution or combination of, neural network architecture(s) for the embeddings and potentially for the BLR data matrix, acquisition function settings, optimization settings. This makes we wonder how easy the approach would be to apply generally. A broader experimental section would help assuage this concern. Originality: The incorporation of sampling distribution embeddings within BO for hyperparameter selection is novel to my knowledge. Quality: The two toy examples are nice proof-of-concepts, but a broader set of experimental results on interesting real-world scenarios would go a long way in providing convincing evidence that the restricted setting and various design decisions aren't issues. The results on the real dataset are positive for the proposed method, but underwhelming in that the easier utilization of a kernel over dataset meta-features (manualGP) and warm-starting with dataset meta-features (initGP) also perform fairly well in comparison. How did the authors make their decisions on the neural network architecture, acquisition function, and optimization settings? Clarity: I don't have major comments here. I found the paper fairly easy to follow. Also, even though I have concern regarding the design space, I would like to point out that the authors have provided a lot of detail on their experimental setup.

Reviewer 2



The paper proposes to transfer information across tasks using learnt representations of training datasets used in those tasks. This results in a joint Gaussian process model on hyperparameters and data representations. The developed method has a faster convergence compared to existing baselines, in some cases requiring only a few evaluations of the target objective. Through experiments, the authors show that it is possible to borrow strength between multiple hyperparameter learning tasks by making use of the similarity between training datasets used in those tasks. This helps developing the new method which finds a favourable setting of hyperparameters in only a few evaluations of the target objective.

Reviewer 3



This paper proposed a novel method for transfer learning in Bayesian hyperparameter optimization based on the theory that the distributions of previously observed datasets contain significant information that should not be ignored during hyperparameter optimization on a new dataset. They propose solutions to compare different datasets through distribution estimation and then combine this information with the classical Bayesian hyperparameter optimization setup. Experiments show that the method outperforms selected baselines. Originality: the method is novel, although it mostly bridges ideas from various fields. Quality: I would like to congratulate the authors on a very well written paper. The quality is very high throughout. Clarity: The paper is clear in general, but left me with some questions. For instance, what criterion is used to learn the NN embeddings of data distributions (Section 4.1)? Is it the marginal likelihood of the distGP/distBLR? Is it learned at the same time as the other parameters mu, nu and sigma? I am not deeply familiar with the Deep Kernel Learning paper but in there it seems they do use marginal likelihood. Significance: I think the paper has the potential to have a high impact in the field. I would have liked more extensive experiments to showcase this, e.g., evaluating on extensive benchmarks such as the ones in [Perrone et al. 2018]. Is there any specific reason why the ADAM optimizer was chosen for the hyperparameters (asking because most of the literature seems to have settled on L-BFGS to get rid of the learning rate)? I will finally mention in passing that the appendix was referenced a total of 14 times in the paper, and while this has no bearing on my final score for the paper, it does feel a bit heavy to have to go back and forth between the paper and the appendix. POST REBUTTAL: I have read the rebuttal, and thank the authors for the clarifications -- it seems I had misunderstood some parts of the paper. I still think this is a decent addition to make to the literature on hyperparameter optimization. I have seen instances of problems where this solution would be applicable.

[Author Response · NeurIPS 2019]

**Reviewer 1, 3, 4** Thank you for all the reviewers time and effort. Regarding the comments on experiments, we would like to emphasise that we have made significant effort into design of experiments in this work. In particular, we have designed one unsupervised example to explain our methodology, as well as 3 additional simulated regression/classification examples to demonstrate the behaviour of our methodology vs that of the baselines. In addition, we have demonstrated our methodology on two real life datasets, Parkinson (Appendix C.5) and a real life protein-ligand binding problem. In the latter, we especially vary the target task to be a different protein, enabling 7 different problems of transfer. To ensure the proposed method applies to different models, we also consider two separate $f$s, namely C-SVM and Random forest (with 5 hyperparameters). In many of these cases, distGP/distBLR was able to outperform its baselines.

Figure 1: Protein with random forest across $140$ evaluations with different NN structure for distGP's $\phi_x$.

**Reviewer 1** Thank you for your detailed review. Regarding the limitation to a single class model and instance space, while we do constrain broad applicability, we believe that there are still many applications in this setting, including ride-sharing, customer analytics model and online inventory system (discussed in [1]). Here, the idea is to re-train our model when new data is available. Furthermore, we demonstrate that our methodology is applicable to a real life protein-ligand binding problem in the area of drug design, which typically require significant efforts to tune hyperparameters of the models for different targets [2].

Here we explain our design space (see additional details in Appendix A.3, B and C); (i) *Choice of embedding (joint vs marginal+conditional)*: We investigate this in Appendix C.1. The results are fairly invariant, as the same information is encoded. (ii) *Acquisition function*: We only investigate the modelling of $f^{\text{target}}$ in this work, and hence we use the standard expected improvement (Appendix A.3). (iii) *Optimisation*: We follow the same regime as in [3] (Section 4.3), but ADAM is used instead, please see reviewer 4 comments below for more information. (iv) *BLR NN structure*: We use the same structure as in [3] for a fair comparison. 5) *Embedding structure*: We choose a small two layer neural network to avoid over-parameterisation, as the number of evaluations, $N$ is small. We also provide new experiments in Figure 1 to show that our methodology is indeed robust to this choice (note that for classification, we use a one hot encoding for $\phi_y$, and that here the difference of accuracy between various NN structures is on order of $0.2\% - 0.3\%$ only).

*initGP and manualGP vs distGP*: While initGP is indeed simple to use, we disagree that its performance is similar to that of distGP, as in many experiments, initGP performance is very poor. Even in the real life protein experiment, its convergence was much slower compared to other methodologies, especially for the case where $f$ is a random forest with 5 hyperparameters in Figure 3 of Section 5. This is to be expected, considering the target task is often different to the source tasks (note hyperparameters are fairly high dimensional here), and that there is no joint model making use of all evaluations. In the case of manualGP, utilisation is not easier than that of distGP, partially as the framework is the same, except that the data embedding is replaced by the manual meta-features (the list is provided in Appendix B.1). Here, there is a design choice of what meta-features to use, and the computation of *landmarker* meta-features can involve substantial effort, as we run simple algorithms to obtain characteristics of the problem (e.g. we use 1-NN classifier, LDA, naive Bayes and decision tree classifier landmarkers in this work for classification). Regarding performance, although distGP does not significantly outperforms manualGP in the real life experiment, we do demonstrate that such use of meta-features can perform poorly in a regression experiment in Section 5. It is also noted that manualGP that can jointly model tasks, while performing feature selection of meta-features is novel in our work.

**Reviewer 3** Thank you for your review, and for comments regarding experiments, please see above.

**Reviewer 4** Thank you for your positive comments regarding the quality of the paper. Regarding your comments on clarity, indeed the criterion to learn the NN embeddings, kernel parameters and other parameters $\mu$, $\nu$ and $\sigma$ is the marginal likelihood of the GP or BLR. All these parameters are learnt simultaneously, in an end-to-end fashion. Regarding extensive benchmarks, we have not used the datasets found in [3], as these datasets' covariate space lies in various $\mathcal{X}$ (i.e. of different dimensions), whilst our work focuses on scenarios where $\mathcal{X}$ is the same across all tasks. For additional comments regarding experiments and application, please see above. We use ADAM as an optimiser, because for each iteration of the training, we use a sub-sample (batch) of the training data to estimate the embedding of distribution, in order to reduce computational cost. Hence, we decided to use ADAM which in general works well with stochastic gradients. In experiments, the loss converges and the model performs well given a reasonable learning rate. However, we agree with the author that L-BFGS can be used an alternative to remove the choice of the learning rate. Finally, thank you for the comment on frequent Appendix referencing - we will make sure to improve the flow.

[1] Matthias Poloczek, et al. "Warm starting Bayesian optimization". pages 770–781. IEEE Press, 2016.
[2] Ross, G.A., et al. "One size does not fit all: the limits of structure-based models in drug discovery." JCTC 9.9 (2013)
[3] Perrone, Valerio, et al. "Scalable hyperparameter transfer learning." NeurIPS. 2018.


[Meta-Review · NeurIPS 2019]

The reviewers overall found the ideas in the paper to be interesting, novel and widely applicable to Bayesian optimization problems.